# Cellulose-Based Nanomaterials Advance Biomedicine: A Review

**DOI:** 10.3390/ijms23105405

**Published:** 2022-05-12

**Authors:** Hani Nasser Abdelhamid, Aji P. Mathew

**Affiliations:** 1Department of Materials and Environmental Chemistry, Stockholm University, SE-10691 Stockholm, Sweden; aji.mathew@mmk.su.se; 2Advanced Multifunctional Materials Laboratory, Department of Chemistry, Faculty of Science, Assiut University, Assiut 71515, Egypt

**Keywords:** cellulose, biomedical, wound healing, drug delivery, antibacterial, tissue engineering

## Abstract

There are various biomaterials, but none fulfills all requirements. Cellulose biopolymers have advanced biomedicine to satisfy high market demand and circumvent many ecological concerns. This review aims to present an overview of cellulose knowledge and technical biomedical applications such as antibacterial agents, antifouling, wound healing, drug delivery, tissue engineering, and bone regeneration. It includes an extensive bibliography of recent research findings from fundamental and applied investigations. Cellulose-based materials are tailorable to obtain suitable chemical, mechanical, and physical properties required for biomedical applications. The chemical structure of cellulose allows modifications and simple conjugation with several materials, including nanoparticles, without tedious efforts. They render the applications cheap, biocompatible, biodegradable, and easy to shape and process.

## 1. Introduction

The word ‘’biomedicine’’ refers to ‘’clinical medicine based on physiology and biochemistry’’ [1]. Biomedicine aims to provide new strategies for biosensing, bioimaging, biomimetic structures, and processes for the regeneration of tissues or organs [2,3,4,5]. It is a multidisciplinary research area involving biologists, chemists, biomaterials scientists, and physicists. An ideal biomaterial for biomedicine should meet specific requirements, including high biocompatibility (no toxicity), high hemocompatibility (no blood coagulation), good biodegradability and bio-absorbability, the promotion of cellular and tissue interactions, the facilitation of an interconnected porous structure, and good mechanical properties such as high wear resistance. Biomaterials should be able to sustain loads and generate tiny wear debris.

Cellulose is a natural linear structural biopolymer of anhydroglucose monomers linked via β-(1–4) bonds ((C_6_H_10_O_5_)_n_; n is the degree of polymerization; n equals 10,000–5000, depending on the source used for cellulose extraction) [6,7,8,9,10,11,12]. Plants, seaweeds, sugarcane bagasse, tunicate, marine algae, and bacteria are predominant sources of cellulose [13,14,15,16]. The annual production of cellulose is more than several hundred billion tons. The market demand has continuously increased over the years. Cellulose exhibits good mechanical, physical, and chemical properties, such as high stability under acidic conditions, chirality, high tensile strength, good elastic modulus (130–150 GPa), low density or lightweight (density of 1.6 g/cm^3^), high biodegradability, and abundant hydroxyl functional groups on their surfaces, enabling chemical modification with high wettability. Thus, cellulose has advanced energy, environmental, and health-based applications [17,18,19,20,21,22,23,24,25,26,27].

Cellulose-based materials include composites with advanced organic and inorganic biomedical applications [28,29,30,31,32,33,34,35,36,37,38,39,40,41,42,43,44]. They have been reported as antibacterial agents [45,46,47,48,49,50] and materials for wound dressing [51,52,53,54,55,56,57], drug delivery [24,58,59,60,61], tissue engineering [33,43,62], artificial blood vessels [63,64], and protection from ultraviolet (UV) radiation [65,66]. Cellulose can be processed into different forms, such as hydrogels [67,68], aerogels [69], membranes [70], and three-dimensional (3D) scaffolds [71,72]. They possess good binding properties [73]. The surface chemistry of cellulose can be modified with several functional groups and compounds [74,75,76,77]. Cellulose-based materials exhibit great potential in biomedicine [78].

This review summarizes the applications of cellulose-based materials in biomedicine. It covers the potential of cellulose and its composites with other materials for drug delivery, tissue engineering, wound healing, antifouling, and antimicrobial agents (Figure 1). Cellulose-based materials exhibit several advantages: high biocompatibility, transparency, production from renewable sources, low cost, high mechanical and physical strength, ease of shaping and processing, and simple procedures for conjugation with other materials.

## 2. Cellulose Nanoparticles

Cellulose has been marketed in several forms, including microfibrillated cellulose (MFC), microcrystalline cellulose (MCC), nanofibrillated cellulose (NFC), cellulose nanocrystals (CNCs), microfibrils, and bacterial cellulose (BC). It can be obtained in micro- and nanoscale regimes (Figure 2). MCC is composed of micro-sized cellulose particles with a length of up to 1 µm and a width of more than 1 µm. MFCs are commonly produced via the mechanical or chemical treatment of wood. Microfibrils are individual fibers with a length of >10 µm and a width of 2–20 µm. At the nanoscale, cellulose nanomaterial or nanocellulose (Figure 2). CNCs, 2,2,6,6-tetramethyl-piperidine-1-oxyl (TEMPO)-mediated oxidized cellulose nanofibers (TOCNFs), and BC are common cellulose nanoparticles. The production method of TOCNFs depends on the oxidation of C_6_ (i.e., -CH_2_OH) of glucose using TEMPO as an oxidant in NaClO_2_ at pH 10. The steric hindrance in TEMPO due to the methyl groups allows only the oxidation of C_6_. CNCs are common products of the acid hydrolysis of cellulose-based materials [79]. The size distribution of CNCs is in the range of 50–500 nm and 2–20 nm in length and width, respectively. BCs are produced by bacterial cells. They exhibit high crystallinity with good elastic and mechanical properties. The length of BC tends to be 200–3000 nm, and the width can be 10–75 nm (Figure 2). Cellulose extraction at the nanoscale removes defects associated with the hierarchical structure and leads to new properties, such as a large surface area.

Cellulose nanomaterials exhibit a degree of crystallinity of 50–90%. The joint functional group of cellulose is the hydroxyl group. NFC can be further oxidized to form carboxylic functional groups via TEMPO-mediated oxidation, resulting in TOCNFs. The cellulose surface can be modified using several methods, including adsorption or chemical modification via covalent bond formation [5]. The surface charge of cellulose is an essential parameter for the material’s characterization. The cellulose colloids’ high charge ensures high stability and prevents nanoparticle aggregation [80].

## 3. Applications of Cellulose Nanoparticles as Antibacterial Agents

Cellulose exhibits no intrinsic biocidal activity, in contrast to other natural biopolymers such as cationic chitosan [81]. However, it can be used as an antibacterial agent via several methods, including surface modification and conjugation with antibacterial agents, including organic and inorganic materials (Figure 3) [47,82]. The following section summarizes the applications of cellulose-based materials as antibacterial agents.

### 3.1. Pure Cellulose for Antibacterial Agents

The antibacterial properties of pure cellulose can be improved via surface modification with several functional groups, such as carboxylic groups, aldehyde, amine, alkylamine, and quaternary ammonium groups (Figure 3). The functional groups of cellulose can be modified with organic molecules that exhibit photosensitization properties. A summary of some of the antibacterial agents prepared using cellulose-based materials is tabulated in Table 1.

The antibacterial activity of 2,3-dialdehyde nanofibrillated cellulose (DANFC) was investigated against *Staphylococcus aureus* (*S. aureus*) and methicillin-resistant *S. aureus* (MRSA) [83]. Dialdehyde formation was performed via the oxidative cleavage of C_2_ and C_3_ bonds in the D-glucose monomer of cellulose using a chemical reagent, such as sodium periodate (NaIO_4_). The antimicrobial activity of DANFC is enhanced by increasing the time of oxidation. The antibacterial effect of DACNF is due to its aldehyde groups, which cause a drop in the pH value (5.7–6.2) [83]. Dialdehyde microcrystalline cellulose (DAMC) antibacterial activity was also reported [84]. DAMC with aldehyde contents of 5.14 mmol/g showed the most potent antibacterial activity against *S. aureus*, *Bacillus subtilis* (*B. subtilis*), *E. coli*, and *Salmonella typhimurium* (*S. Typhimurium*). It exhibited minimum inhibitory concentration (MIC) values of 15, 15, 15, and 30 mg/mL for *S. aureus*, *B. subtilis*, *E. coli*, and *S. Typhimurium*, respectively [84]. Cellulose was extracted from ginger residues, denoted as GNFs (ginger nanofibers), via acid hydrolysis and high-pressure homogenization [85]. GNFs were tested for antibacterial activity [85]. The MIC values of GNFs were 14 ± 2, 13 ± 1, 18 ± 0, and 31 ± 0 µg/mL for *B. cereus, E. coli*, *S. aureus*, and *S. Typhimurium,* respectively [85].

Cellulose with carboxylic groups shows high antibacterial activity (Table 1). A gel of TOCNFs (0.2–0.8 wt.% in water) inhibited the growth of a wound infected with the pathogen *P. aeruginosa* [86]. The physical, chemical, and antibacterial activities of carboxylated CNFs can be modified via treatments such as autoclaving [87]. Autoclave treatment at 121 °C for 20 min reduced the gels’ viscosity, increased the ultraviolet-visible absorbance maxima to 250 nm, and increased the aldehyde content. Autoclaved carboxylated CNFs showed high antibacterial activity with minimal toxicity toward L929 mouse fibroblasts and reconstructed human epidermis (RhE) [87]. The antibacterial activity of carboxylated CNFs with different oxidation levels was evaluated against *P. aeruginosa* and *S. aureus* [55,87]. Oxygenated CNFs exhibited higher antibacterial activity than non-oxygenated CNF dispersion [55,87].

The antibacterial activity of pure cellulose nanoparticles could be due to several mechanisms, including a decrease in the mobility of bacterial cells [86], the surrounding and entrapping of bacteria via the formation of a network [55], and a reduced pH value due to the increase in aldehyde groups in CNFs [83,87]. However, further investigations should be carried out to understand the key parameters affecting the antibacterial activity of pure cellulose nanomaterials.

Pure cellulose with suitable functional groups possesses good antibacterial activity. However, it is crucial to consider the presence of foreign species such as endotoxins or lipopolysaccharides that can cause inflammation leading to antibacterial activity [88]. A modified TEMPO-mediated oxidation method using sodium hydroxide as a pre-treatment produced CNFs showing an endotoxin level of 45 endotoxin units (EU) per gram of cellulose [89]. This value may be non-toxic at low concentrations. However, it can be critical at high concentrations [90]. The presence of endotoxin should be considered during the evaluation of the antibacterial activity of cellulose-based materials.

### 3.2. Photoactive Cellulose for Antibacterial Agents

Photo-based light radiation treatments are promising for antibiotic-resistant bacteria (Table 1) [91,92,93,94]. They require the presence of a photosensitizer molecule that converts the light radiation to thermal energy (photothermal therapy) or generates reactive species (i.e., photodynamic treatment) such as reactive oxygen species (ROS). Pure cellulose lacks photosensitizer properties. Thus, it is usually modified with small molecules via covalent or non-covalent interactions to absorb light. Most of these photosensitizers are bacteria-inactivating materials. However, they are effective for bacterial inactivation using cheap light sources such as light-emitting diodes (LEDs) [95].

Photodynamical inactivation (PDI) of bacteria using CNC [96] and hairy aminated nanocrystalline cellulose (ANCC) was reported [97]. ROS can be generated under light via the modification of cellulose with molecules such as anthraquinone vat dyes [98], 3,3′,4,4′-benzophenone tetracarboxylic acid [99], ketoprofen [100], hypocrellin [101], xanthene [77], BODIPY (**Dipy**rromethene **bo**ron difluoride) [91,102], chlorin-e6 [103], phthalocyanines [104,105], protoporphyrin-IX [106,107,108], and porphyrin [96,97,109,110,111,112,113,114]. CNC was chemically modified with cationic porphyrin, denoted as CNC-Por, via Cu(I)-catalyzed Huisgen–Meldal–Sharpless 1,3-dipolar cycloaddition (Figure 4a). The reaction occurs between the azide and alkyne groups on the cellulosic and porphyrinic molecules, respectively (Figure 4a). PDI of *Mycobacterium smegmatis, S. aureus*, and *E. coli* by CNC-Por was investigated under white-light radiation (400–700 nm, 60 mW/cm^2^) [96]. CNC-Por exhibited high PDI against *M. smegmatis* and *S. aureus* and insignificant activity against *E. coli* after illumination for 15 min. However, after 60 min, the PDI activity of the material against all bacteria was higher than 99% (99.9999% for *S. aureus*) [96]. ANCC was modified with the natural photosensitizer Rose Bengal (RB) via covalent bonds (Figure 4b) [97]. RB-ANCC showed PDI of over 80% against the pathogens *Listeria monocytogenes* and *S. Typhimurium* under illumination using normal light irradiation. Interestedly, ANCC improved the PDI of free RB against *S. Typhimurium* [97].

Cationic porphyrin (Por(+))-conjugated cellulose was formed as paper for scalable antimicrobial treatment using PDI [81]. Por(+)- modified cellulosic paper was illuminated using visible light with a wavelength and power of 400–700 nm and 65 ±5 mW/cm^2^, respectively, for 30 min. The antibacterial and antiviral efficacies were investigated against bacteria (*S. aureus*, *vancomycin-resistant Enterococcus faecium (VER)*, *Acinetobacter baumannii*, *P. aeruginosa*, and *Klebsiella pneumoniae*) and viruses such as dengue-1 virus, influenza A, and human adenovirus-5 with efficiencies of 99.995%, 99.5%, and 99%, respectively [91]. The inactivation efficiencies for all investigated species, i.e., bacteria and viruses, were higher than 99.9% [91].

PDI using cellulose-based materials exhibits several advantages, such as high antibacterial efficiency (≈100%, Table 1). The method can be applied to the treatment of antibiotic-resistant bacteria. Cellulose chemistry allows the fabrication of materials such as paper [91], fibers [110], or textiles [115] with scalable and straightforward uses for antibacterial treatment. It allows immediate covalent modification using advanced methods such as photo-strain-triggered click ligation [116]. It may open a new avenue for photoactive textiles [117].

### 3.3. Cationic Cellulose for Antibacterial Agents

Similar to chitosan (CTS), cationic cellulose exhibits intrinsic antibacterial activity. The mechanism of the antibacterial action of cationic biopolymers depends on the high binding affinity between the positive charge of these polymers and the negative charge of bacterial cells. The surface of bacterial cells, i.e., Gram-positive and Gram-negative, is negative due to hostile phosphate groups in peptidoglycan and phospholipids. Following this principle, creating a positive charge on cellulose enables high antibacterial activity (Table 1). Cationic CNCs can also be used as immune modulators [118].

Cationic cellulose can be achieved via modification with quaternary ammonium compounds such as poly(isopropanol dimethyl ammonium) chloride (PIDMAC) [119], quaternized poly (2-(dimethyl aminoethyl) methacrylate) (PDMAEMA) [120], cetyltrimethyl ammonium bromide (CTAB) [121], 3-chloro-2-hydroxypropyl-tri-methyl ammonium chloride 3-chloro-2-hydroxypropyl-trimethyl ammonium chloride (CHPTAC) [122], pyridinium/N-chloramine [123], benzalkonium chloride [124], and quinolinium silane salt [125]. Cellulose grafted with DMAEMA can be prepared via polymerization using a reversible addition-fragmentation chain transfer (RAFT) reaction [120]. Cationic cellulose can also be prepared via direct covalent bonding of quaternary ammonium moieties without a linker [126]. The silane group of 3-(trimethoxy silyl)-propyl-dimethyl octadecyl ammonium chloride reacted with the hydroxyl functional groups in cellulose, forming Si–O–Si bonds [126]. Cellulose-QA showed complete inactivation of *E. coli* and *P. aeruginosa* after one hour and 10-fold inactivation of *B. cereus* [126].

Cellulose can be modified with quaternary ammonium and porphyrin moieties via esterification [108]. The material’s treatment requires relatively low porphyrin concentrations and can be applied under a low dosage of white-light irradiation (2.4 J/cm) [103]. The protoporphyrin IX (PpIX) moiety provides white-light radiation photosensitization (Figure 5). Quaternary ammonium moieties offer dual functionality: antibacterial activity and the prevention of the aggregation of porphyrins that cause the quenching of ROS generation. Photodynamic treatment of bacteria using quaternary ammonium-porphyrin-modified cellulose showed effective antibacterial activity against antibiotic-resistant *E. coli* and *S. aureus* strains (Figure 5). The antibacterial activity of the materials is due to the intrinsic bioactivity of quaternary ammonium moieties and the generation of ROS [108].

The antibacterial activity of cationic cellulose can be due to several mechanisms, such as destabilization of the bacterial intercellular membranes due to Ca^2+^ or Mg^2+^ ion exchange, membrane disruption because of the release of potassium (K^+^) ions, the formation of ROS, an increase in the number amine groups [127], or an increase in lipophilicity using amino-alkyl [128].

### 3.4. Organic-Modified Cellulose as Antibacterial Agents

The antibacterial activity of cellulose can be enhanced via chemical modification with organic bioactive molecules, including antibiotics, antimicrobial peptides, *N*-halamines, aminoalkyl groups, bacteriophages, and polymers (Figure 3).

#### 3.4.1. Antibiotic-Modified Cellulose

Antibiotics are widely used for bacterial treatments. Cellulose has been grafted with antibiotics, including β-lactam antibiotic benzylpenicillin [129], ciprofloxacin [130], tetracycline hydrochloride [131], silver sulfadiazine (Ag SD) [132], 3-pentadactylphenol [133], allicin [134], and amoxicillin [135]. Antibiotics such as penicillin can be covalently modified with cellulose via ester bond formation (Figure 6) [129]. The covalent modification of cellulose ensures high durability with good antibacterial activity. Using thermal treatment, the materials can be processed into the film without destroying the formed bonds (Figure 6).

The cationic cellulose filter paper was reported for water treatment with antibacterial activity [119]. The cellulose filter paper was coated with PIDMAC as a cationic polyelectrolyte binder (CPE). It was loaded with amphiphilic block copolymer micelles containing triclosan (antibacterial and antifungal agent). The micelles interacted with CPE via polystyrene-block-polyacrylic acid (PS-b-PAA) as the block copolymer. The materials included two antibacterial agents i.e., triclosan (hydrophobic) and an ammonium compound [119]. Thus, they showed high antibacterial activity [119].

Cellulose is good support for antibacterial agents, including antibiotics [136]. The use of conventional antibacterial agents such as antibiotics ensures high antibacterial activity for cellulose-based materials. However, the release of antibiotics into water causes environmental concerns and raises the risk of highly antibiotic-resistant bacteria.

#### 3.4.2. Aminoalkyl-Modified Cellulose

Cellulose modified with aminoalkyl functional groups exhibits antibacterial activity [128,137,138,139,140]. The modification takes place via a reaction with silanol groups as coupling agents. The process involves the formation of covalent bonds between silanol groups (Si-OH) and hydroxyl groups (OH) of cellulose, i.e., the formation of Si–O–C bonds. BC membranes modified with aminoalkyl groups using APMS ((3-aminopropyl)trimethoxysilane) exhibit antibacterial activity against *S. aureus* and *E. coli* [128]. The antibacterial activity of APMS-modified cellulose is due to the polycationic nature of the membrane and the alkyl chains in APMS. The long chains of the alkyl groups increase lipophilicity, which ensures strong interactions with the cytoplasmic membrane of the investigated bacterial cells [128]. Alkyl groups with up to 10 carbons exhibited a broad spectrum of antibacterial and antifungal activities [141]. A cellulose membrane with aminoalkyl groups showed non-toxic properties in human adipose-derived mesenchymal stem cells (hAMSC) [128] and human embryonic kidney 293 cells (HEK-293) [141].

Besides the chain length, the increase in amine groups ensures high antibacterial activity [127]. Cellulose was modified with aminoalkyl containing different groups of 1, 2, and 3 amines using APMS, 2-aminoethyl 3-aminopropyl trimethoxysilane (DAMS), and 3-2-(2-aminoethyl amino) ethyl aminopropyl trimethoxysilane (TAMS), respectively. The increase in the number of amine groups results in high antibacterial activity. Cellulose–TAMS showed the most increased antibacterial activity against Gram-positive bacteria [127].

#### 3.4.3. *N*-halamine@Cellulose

*N*-halamines, halogen atoms linked to nitrogen-containing compounds, modified cellulose [142,143,144]. The functional groups of *N*-halamine, such as epoxy groups or organosiloxane, enable grafting onto cellulose via a reaction with hydroxyl groups [145]. The materials exhibit durable antibacterial activity [145]. The *N*-halamine of the s-triazine-based quaternized molecule was used to modify cellulose. The material showed a 6-log reduction in *S. aureus* and *E. coli* after treatment for 1–5 min [144]. The oxidant chlorine in *N*-halamine molecules showed 50% retention of its activity even after 50 cycles of washing and 30 days of storage. The remaining 50% of the material’s activity could be regenerated after exposure to a bleach solution [144]. *N*-halamine-modified cellulose can be restored via simple methods, such as treatment with a diluted bleach solution [146].

#### 3.4.4. Antimicrobial Peptide-Modified Cellulose

Antimicrobial peptides are attractive compared to small organic molecules used as antibiotics [147,148]. Cellulose has been modified with several peptides, such as gentamicin [149]. Bacterial cellulose (BC) was chemically modified with RGDC peptide (R, G, D, and C refer to arginine, glycine, aspartic acid, and cysteine, respectively) and gentamicin via covalent bonds [149]. The process involves the cross-coupling of RGDC peptide to BC using a coupling agent such as 3-aminopropyltriethoxysilane (APTES). Gentamicin was then attached to the surface of the RGDC-BC membrane [149]. The gentamicin–RGDC-grafted BC membranes exhibited high antibacterial activity against *Streptococcus mutans* without significant toxicity toward human dermal fibroblasts [149].

#### 3.4.5. Polymer-Modified Cellulose

Cellulose has been modified with polymers to enhance antibacterial activity. Polymers can be directly blended (anchored or grafted) with cellulose using a simple procedure (post-synthetic procedure) or via in situ polymerizations of the monomers in the presence of cellulose (in situ procedure). Cellulose was modified with several polymers, including: (1) biopolymers such as chitosan [150] and (2) synthetic polymers such as polypyrrole [151], polypropylene [133,152], polyethyleneimine [153], polyhexamethylene guanidine hydrochloride (PHMG-Cl) [154], polyvinyl [155], 2-aminoethyl methacrylate [156], and poly(3-hydroxy-acetylthioalkanoate-co-3-hydroxy alkanoates) [157].

Fibers of dialdehyde cellulose (DAC)/CTS composite were prepared via stirring and filtration [150]. The antimicrobial activity of DAC/CTS against *E. coli* and *S. aureus* was investigated. Data analysis showed antibacterial activity of 90.2% and 95.1% against *E*. *coli* and *S*. *aureus*, respectively [150]. The antibacterial activity is due to the aldehyde groups of cellulose and the intrinsic antibacterial activity of CTS [150].

Polyrhodanine (PR) was prepared in the presence of CNC [158]. The synthesis procedure involves the polymerization of rhodanine on the surface of CNC using ferric chloride (FeCl_3_) as the initiator and oxidant. The negatively charged surface of CNC assisted the polymerization and led to the formation of core-sheath nanoparticles of CNC@PR. Using the plate colony-counting method, the antibacterial activities were evaluated against *E. coli* and *B. subtilis*. CNC@PR showed good antibacterial activity with good MICs [158].

Cellulose has been modified with polymers such as guanidine polymer [159]. In situ polymerization of 2-aminoethyl methacrylate (AEM) into a BC network was reported with and without a cross-linker such as *N*,*N*-methylenbis(acrylamide) (MBA) [156]. The polymerization took place on the BC network via a radical-based reaction, resulting in BC/poly-AEM/MBA. The synthesized polymer filled the pores of the BC network. BC/poly-AEM and BC/poly-AEM/MBA materials were evaluated for antimicrobial activity against *E. coli*. BC/poly-AEM exhibited higher antibacterial activity than cross-linked BC/poly-AEM/MBA material. The authors explained that cross-linking reduced the diffusion of the bacteria into the BC network, leading to low contact between *E. coli* and the ammonium groups [156]. Thus, cross-linking of the network reduced the antibacterial activity [156].

Post-synthetic modification of BC with octenidine dihydrochloride was reported [160]. The material showed significant antimicrobial activity against *S. aureus* even after six months of storage. It exhibited minimal cytotoxic effects against human keratinocytes [160].

#### 3.4.6. Bacteriophage-Modified Cellulose

Cellulose can be modified with a virus form called a bacteriophage (or phage for a simple description) to induce DNA or RNA inside cells [161,162,163,164] or bacteriophage endolysins (enzymes that cause hydrolysis) [165,166]. The phage can be immobilized on cellulose via non-specific interactions (adsorption) [167] or covalent bonds [168].

Several enzymes, such as lysozyme (muramidase), can be used as antibacterial agents. The mechanism of most enzymes is the cleavage of bonds in the cell membrane, such as the peptidoglycan layer of bacteria, leading to cell lysis [169]. Enzymes such as lysozyme cause hydrolysis of the 1,4-β-linkage between *N*-acetylmuramic acid and *N*-acetylglucosamine [169]. CNC was used to immobilize hen egg-white lysozyme (HEWL) and T4 lysozyme (T4L, Figure 7) [168]. The immobilization of lysozyme on CNC does not cause a decrease in the enzyme’s enzymatic activity for lysis and hydrolysis. The process involves covalent coupling via carbodiimide-activated CNC via carboxylate groups to glutaraldehyde-activated aminated CNC (Am-CNC, Figure 7). Am-CNC-HEWL and Am-CNC-T4L showed lytic activity of 86.3% and 78.3%, respectively. After immobilization, the enzyme exhibited high bactericidal activity compared to the free enzyme. They also showed high stability during storage at 4 °C and 22 °C [168]. A thin film of CNC–lysozyme composite was prepared using the evaporation-induced self-assembly method [170]. The technique produces a film with a HEWL enzyme loading of 10 wt.%. However, it showed insignificant antibacterial activity against Gram-positive bacteria. The optimization of enzyme-modified cellulose is critical to achieving high antibacterial activity.

Enzyme-based antibacterial agents offer several advantages, such as selectivity. For example, lysozyme exhibits higher activity against Gram-positive bacteria than Gram-negative bacteria. This selectivity is because the peptidoglycan layer exists only in Gram-positive bacterial cells [20]. Antibacterial agents should be active against a broad spectrum of bacteria.

Bacteriophages exhibit high antibacterial activity. However, they are usually negatively charged at their heads. Thus, they have difficulty interacting directly with the negative charge of cellulose. They interact easily with the positively charged surface via electrostatic interactions. The presence of functional groups such as amine and carboxylic acid on the phage surface enables functionalization via covalent bonds. The latter method allows highly durable and robust phage attachment to cellulose materials.

Cellulose–organic composites with antibacterial activity can be synthesized via several methods. The simple attachment of both components via grafting or non-covalent interactions requires no tedious efforts or expensive equipment. However, they lack high stability or long-term activity due to leaching. Thus, strong bonds such as covalent bonds are required to ensure high strength and long-term activity [171].

### 3.5. Cellulose–Inorganic Nanoparticles for Antibacterial Agents

Cellulose has been modified with inorganic nanoparticles such as carbon nanomaterials, metal oxides, metallic nanoparticles, and metal-organic frameworks (MOFs) (Figure 3) [172,173,174,175,176,177]. Most of these materials exhibit intrinsic antibacterial activity, leading to high performance against Gram-positive and Gram-negative strains.

Carbon nanomaterials exhibit high antibacterial activity. They were conjugated with cellulose. A composite of BC and graphene oxide (GO) was reported as an antimicrobial agent against *E. coli* and *S. aureus* [178]. Electrostatic modification improved the antimicrobial activity of GO/BC nanocomposites [179]. The antibacterial activity of carbon nanosheets such as GO is mainly due to the sharp edges of the sheets, which act as a knife to cut the cell membrane, causing the rupture of the outer envelope [180,181,182,183].

Nanoparticles with metal oxides such as zinc oxide (ZnO) exhibit high antibacterial activity. The antibacterial activity of ZnO-based nanomaterials depends on the ZnO contents in composites [175]. A composite of cellulose acetate (CA) and hydrophobic polysulfone (PSf) polymer was used to modify the material with 0.1 wt.% ZnO NPs [173]. Membranes containing ZnO NPs showed good antibacterial activity against *E. coli* [173]. ZnO nanoparticles were prepared via an in situ procedure [174]. Zn^2+^ ions were adsorbed onto cellulose before precipitation using ammonium hydroxide [174]. The surface functional groups of cellulose, such as hydroxyl groups, enable the adsorption of Zn^2+^ ions via electrostatic interactions and control the morphology and particle size.

Metal oxide nanoparticles such as CuO [184] and TiO_2_ nanoparticles [185,186], faujasite [187], and montmorillonite (MMT) [188] were modified with cellulose for antibacterial activity. Cellulose acetate/TiO_2_ nanoparticles exhibited high antibacterial activity [186]. A faujasite–cellulose composite membrane was used to purify water contaminated with bacteria (*E. coli*, *Enterococci*, and *Clostridium*). It showed high removal efficiency, resulting in <100 colonies/100 mL [187]. The presence of MMT in the cellulose membrane enabled modification with several metal ions, such as Na, Ca, and Cu [188]. BC/Cu-MMT composites showed the highest antibacterial activity against the investigated bacteria [188].

Silver nanoparticles (Ag NPs) are very active antibacterial agents [189]. They have been widely modified with cellulose nanoparticles for antibacterial activity [190,191,192,193,194,195,196,197,198]. Ag NPs were synthesized directly into electrospinning fiber cellulose acetate (CA) [199]. The procedure involved the adsorption of silver ions reduced via photons onto Ag NPs. The prepared materials exhibited high antibacterial activity against *S. aureus*, *E. coli*, *K. pneumoniae*, and *P. aeruginosa* [199]. A nanofiltration (NF) membrane consisting of CNC/silver (CNC/Ag) and polyamide was fabricated using the interfacial polymerization (IP) method [200]. The membrane with 0.01 wt.% CNC/Ag showed antibacterial activity of 99.4% against *E. coli* viability [200]. The high antibacterial activity of silver-based nanomaterials can be explained by several mechanisms, such as ionization and the release of silver ions [198]. Silver nanoparticles offer several advantages. They can be synthesized into cellulose textiles via an in situ procedure [201]. They can be conjugated with other materials such as magnetic nanoparticles (MNPs) [202] and gold nanoparticles [203]. Furthermore, cellulose serves as a biodegradable support for Ag NPs [204].

The conjugation of cellulose with inorganic nanomaterials ensures high antibacterial activity. Most of the antibacterial activity is due to the inorganic materials present in the cellulose composite [205]. However, they can be toxic to mammalian cells or environmentally unfriendly due to toxic heavy metal ions released into drinking water.

## 4. Cellulose-Based Materials for Antifouling

Pure cellulose nanoparticles with residual lignin [206,207] or carboxylic functional groups [208] possess antifouling properties. Thus, they are widely used for membranes for antifouling applications [208]. Cellulose nanoparticle-based membranes exhibit high antifouling performance with high reflux compared to commercial membranes such as Millipore GS9035 [209]. A membrane of TOCNF, poly(vinyl alcohol) (PVA), and polyethersulfone (PES) was fabricated for antifouling application. The TOCNF/PVA@PES membrane exhibited high antifouling capability [208]. A micro/nanocellulose membrane grafted with zwitterionic poly(cysteine methacrylate) (PCysMA) showed excellent antibacterial and antifouling properties. It showed a reduction efficiency of 85% in the biofilm formation of *S. aureus* [207].

Photocatalytic disinfection using photoactive substances is based on generating free radicals under a light. A CNC/silver/polyamide membrane containing CNC/Ag (0.01 wt.%) showed high antifouling activity of 92.6% using humic acid with the antibacterial activity of 99.4% against *E. coli* viability [200]. A membrane of CMC/GO/magnesium oxide (MgO) nanoparticles (CMC/GO/MgO) was prepared for photocatalytic antifouling [210]. It can be used as an antifouling membrane due to the generation of electrons and ROS. Thus, it can be used to oxidize organic pollutants [210].

Cellulose–inorganic hybrids exhibited high antibacterial performance. CNC/silver/polyamide membrane demonstrated effective antifouling properties using humic acid with a flux recovery of 92.6% [200]. Composite nanofibers of CA and polysulfone (PSf) with 0.1 wt.% ZnO exhibited high antibacterial activity against *E. coli* [211].

**Table 1 ijms-23-05405-t001:** Antibacterial applications for cellulose-based materials.

Materials	Preparation Methods	Form	Microorganism	Methods	Time	Efficiency	Mechanism	Ref.
DANFC	1. Mechanical grinding 2. Enzyme treatment 3. NaIO_4_ oxidation 4. Dialysis 5. Freeze-drying	Mat	*S. aureus MERSA*	Zone inhibition Plate counting method	24 h	100%	Drop-in pH value	[83]
TOCNF	1. TEMPO oxidation 2. Oxygenation	Suspension	*P. aeruginosa* *S. aureus*	Plate counting method	24 h	71%	The formation of a network surrounding the bacteria	[55]
	1. TEMPO oxidation 2. Autoclaving (121 °C, 20 min)	Gels			24 h	71%	Increase in aldehydes, drop in pH value	[87]
CNC-Porphyrin	1. HBr acid hydrolysis of Whatman filter paper 2. Cu(I)-catalyzed Huisgen–Meldal–Sharpless 1,3-dipolar cycloaddition	Suspension	*Mycobacterium smegmatis*, *S. aureus*, *E. coli*		60 min	>99%	Generation of ROS, photodynamic	[96]
Porphyrin-cellulose paper	Cu(I)-catalyzed Huisgen–Meldal–Sharpless cycloaddition	Paper	*S. aureus*, *VER*, *Enterococcus faecium*, *Acinetobacter baumannii*, *P. aeruginosa*, *Klebsiella pneumoniae*		30 min	>99.9%		[91]
NFC-Porphyrin	Cyanuric chloride coupling	Paper	MRSA, VER, *E. faecium*, *A. baumannii* *K. pneumoniae*		30 min	99.999%		[114]
CHPTAC-Cellulose triacetate	1. Immersion precipitation technique 2. Alkaline hydrolysis 3. Esterification	Membrane	*S. aureus* *E. coli*		24 h	78.7–89.0% 64.7–76.6%	Cationic charge	[122]
BC-aminoalkyl	Stirring for 5 h at 25 °C			Dynamic shake flask method	24 h	>99.9%	Increase in lipophilicity	[128]
DAC/CTS	1. NaIO_4_ oxidation 2. Stirring 3. Filtration	Fibers		Plate counting method	24 h	95.1% 90.2%	Drop in pH, cationic CTS	[150]
T7 phage-Cellulose acetate	Electrospinning of cellulose acetate	Membrane	*E. coli*	Plaque-forming units (PFU)	24 h	6 log(PFU/mL)	Release of phage and hydrolysis	[167]
Am-CNC-HEWL Am-CNC-T4L	1. Ammonium persulfate oxidation 2. Modification 3. Coupling	Suspension	*M. lysodeikticus*, *Corynebacterium* sp., *E. coli*, *P. smendocina*	Time–kill study with Alamar Blue assay	24 h	100%	Lytic activity	[168]
CNC-Lysozyme	1. Sulfuric acid hydrolysis 2. Evaporation-induced self-assembly	Thin film	*E. coli* *S. aureus*	Diffusion assays	24 h	0%		[170]
ZnO-BC	In situ synthesis	Sheets		Inhibition zone	24 h	5.7 ± 0.29 mm 2.9 ± 0.75 mm	Formation of ROS	[212]
	Ex situ synthesis	Film	*E. coli*	Inhibition zone	24 h	34–41 mm		[213]
TiO_2_-BC	Ex situ synthesis	Pieces		Optical density (OD_600_)	12 h	60%		[185]
Faujasite-cellulose	Hydrothermal	Membrane	*E. coli*, *Enterococci* *Clostridium*	Standard FS ISO9308 and ISO17025		80%		[187]
SSD/BC	Impregnation and ultrasonication	Commercial membrane	*P. aeruginosa*, *E. coli* *S. aureus*	Zone inhibition	24 h	6.5 mm		[214]

Inorganic nanoparticles improved the antibacterial activity of cellulose-based materials. Direct contact with metallic ions causes damage to the cell membrane due to the generation of ROS that can permeate the cell wall. The formed ROS damage the phospholipid layer of the cell membrane and can also cause cell disruption via the oxidation of thiol groups of amino acids present in the bacterial cells.

## 5. Cellulose Nanoparticles for Wound Dressing

During burn treatment, special care should be taken to avoid microbial infection [55]. Wound healing requires fast re-epithelialization within 10–14 days to prevent complications. Chronic wounds may undergo critical physiological changes or initiate tumor growth, leading to the significant destruction of tissues or organs. This situation is vital for diabetic patients who suffer from chronic ulcers. Thus, the dressing becomes an essential active component during the healing process and is no longer considered a supplement. The sauce should offer a warm and moist environment to provide a fast and effective healing process. It should also be biocompatible and easy to detach, act as a thermal insulator, and provide antimicrobial activity to prevent infections.

Natural polymers such as cellulose have been widely investigated for wound dressing because of their high biocompatibility and biodegradability (Table 2). Bacterial cellulose (BC) has been extensively used for wound healing [57,215]. Several methods have been reported to modify BC to improve the material’s properties, such as biocompatibility and the uptake/release of water or drug molecules, and ensure high antimicrobial activity. Cellulose can be fabricated into different forms, such as films, foams, hydrocolloids, hydrogels, and nonwovens [216]. It can also supply wounds with drugs, growth factors, and antimicrobial agents. The traditional gauze of cotton is usually modified with sodium periodate via oxidation, i.e., the formation of dialdehyde [217]. The oxidation process decreases elastase and can support the healing of a chronic wound. It can also improve the antibacterial activity and offers high protection due to the drop in pH value caused by the aldehyde functional group in cellulose [83]. BC-based wound dressings reduce the closure time (i.e., recovery rate) of the wound [218,219] without significant inflammation (Table 2) [220].

A wound dressing of pure cellulose such as NFC was reported [53]. The antimicrobial properties of dressing materials were investigated against bacterial pathogens such as *S. aureus* and *P. aeruginosa* [53]. NFC-based wound dressing showed no antibacterial activity against the tested strains. The BC-based film exhibits good wound healing (Figure 8) [220]. The performance is lower than that of traditional wound dressing (gauze). However, it may be improved. The sides of BC film (i.e., bottom and top) exhibit different performances (Figure 8) [220]. The antibacterial activity of cellulose can be improved via carboxylated CNFs [55]. Autoclaved CNFs showed a strong antibacterial effect against bacteria that infect wounds [87].

Cellulose nanofibrils were prepared via the defibrillation of wood via a mechanical method [51]. The fibrils were developed into a membrane via filtration and drying under mild pressure. The prepared cellulose membrane was used as a wound dressing. It offered strong adhesion to the wound and promoted epithelialization after four days without the observation of allergic reactions or inflammation during the treatment [51]. A cellulose hydrogel was prepared from the alkali treatment and TEMPO-mediated oxidation of ginger fibers (T-GNFs) [221]. It was designed via a simple vacuum-assisted filtration using ginger nanofibers without cross-linking treatments. The preparation method required low energy and few components for the production procedure. The material’s antibacterial and wound healing properties were investigated [221]. The material enabled the closure of the wound without significant antibacterial activity against *E. coli* and *S. aureus* [221]. The 3D printing of NC hydrogel scaffolds was reported for wound healing [222]. The process included a double cross-linking procedure via two steps: (i) in situ Ca^2+^ cross-linking during 3D printing and (ii) chemical cross-linking with 1,4-butanediol diglycidyl ether (BDDE) after printing. The scaffolds were used to support the proliferation of fibroblast cells with an attachment percentage of 84–86.5% [222]. Cellulose exhibits high performance in wound healing and can be processed via several procedures.

Modulating the wound’s pH value enhanced the wound treatment [223]. Intact skin exhibits a slightly acidic pH value of 4–6 due to the secretion of organic acids by keratinocytes to regulate bacterial flora and prevent infection, while the infected wound exhibits a pH value of 7–7.5. The neutral pH value of the injury is an ideal condition for the growth of bacterial cells. The drop in the pH of the wound using aldehyde-modified cellulose enabled high protection against bacterial infection [83]. Dialdehyde of microcrystalline cellulose (DAMC) [84] and nanofibrillated cellulose (DANFC) [83] with aldehyde contents of 6.5 mmol/g and 1.5 mmol/g, respectively, showed high antibacterial activity.

The antibacterial activity of a cellulose-based wound dressing can be improved by loading it with antibiotics. Chloramphenicol (CAP)-loaded dialdehyde (DABC) and non-oxidized BC membranes were investigated in terms of their antimicrobial efficiency against *E. coli*, *S. aureus*, and *Streptococcus pneumoniae* (*S. pneumoniae*) [224]. The CAP drug-loading capacity using DABC was low (0.1 mg/cm^2^) than the BC-based membrane, which showed a loading capacity of 5 mg/cm^2^. However, both membranes, i.e., oxidized and non-oxidized BC, showed high antimicrobial activity against the tested bacteria. CAP/DABC showed high adhesion and proliferation of the fibroblast cell line L929 compared to non-oxidized BC. This study highlighted the potential for using newly developed CAP/DABC dressing materials in wound treatment [224].

Cellulose-modified organic nanocomposite exhibits high performance as a wound dressing. A three-dimensional (3D) network of collagen I (Col-I), hydroxypropyl trimethyl ammonium chloride chitosan (HACC), and BC was prepared via a membrane–liquid interface (MLI) method [225]. HACC exerted antibacterial activity during wound healing [225]. Inerpan (a polymer of L-leucine and methyl L-glutamate) and Procel-Super (SOD) accelerated the healing of burn wounds with BC-based wound dressing by 17.0 and 5.5%, respectively [226].

Cellulose nanocomposites have been widely reported for wound dressing. A membrane of BC and chitosan (BC–CTS) was fabricated via the immersion of BC in a solution of CTS followed by freeze-drying [227]. The procedure can be applied for large-scale production. The presence of antibacterial agents such as CTS significantly inhibited the growth of bacterial cells, e.g., *E. coli* and *S. aureus* [227]. The BC-CTS membrane exhibited higher antibacterial activity compared to the pure BC membrane. BC and BC-CTS membranes demonstrated antibacterial inhibition of 49.2% and 99.9% against *E. coli* and 30.4% and 99.9% against *S. aureus*, respectively. The BC-CTS membrane showed high epithelialization and regeneration during wound healing compared to wounds treated with BC only or a commercial dressing such as Tegaderm^TM^ [227].

A sponge of carboxylated brown algae cellulose nanofibers (BACNFs) was prepared to contain organic rectorite (OREC) via freeze-drying [228]. OREC was organized via intercalation of chitosan (positively charged) into negative silicate layers via ion exchange. The sponge of BACNFs/OREC exhibited high antibacterial activity against *E. coli* and *S. aureus* without significant toxicity toward mouse fibroblasts (L929). It also showed effective wound closure (100%) after 12 days compared to commercial gauze. It also exhibited effective prevention of bacterial infections without a significant inflammatory response [228]. It is essential to mention that BACNFs showed no antibacterial activity [228].

Cellulose has been conjugated with several inorganic-based antimicrobial agents, such as silver nanoparticles [229], graphene oxide (GO) [230], and ZnO [231]. Silver-based drugs such as silver sulfadiazine (SSD) are widely used as antibacterial agents. BC/SSD was prepared via the impregnation of a BC membrane with SSD via ultrasonication [214]. The BC/SSD membrane showed significant antibacterial activities against several bacteria, such as *P. aeruginosa*, *E. coli*, and *S. aureus* [214]. The membrane exhibited high biocompatibility [214]. Methyl cellulose improved the dispersion of GO [230]. A cytocompatibility assay using [3-(4,5-dimethylthiazol)-2-yl]-2,5-diphenyltetrazolium bromide (MTT) and live/dead assays of EA.hy926 human endothelial cells (ECs) exhibited high biocompatibility. Cell migration under the effect of GO–cellulose using an induced wound scratch model of EA.hy926 ECs was reported (Figure 9A). GO–cellulose accelerated cell migration. The in vivo wound healing of full-thickness wounds in the dorsum of rats (diameter of 8 mm) was evaluated using different measurements, including wound closure measurements (Figure 9B,C). GO–cellulose exhibited a significant wound closure rate compared to control experiments (Figure 9B,C). Data analysis revealed high wound re-epithelization and neovascularization with the significant formation of extensive collagen deposition [230].

Cellulose offers several advantages for wound dressing (Table 2). The required cost is low. It can be fabricated into applicable forms such as membranes using cheap sources such as wood [51]. A cellulose-based membrane exhibited higher performance as a wound dressing than a commercial porous regenerating membrane [51]. The epithelialization of wood-based dressings such as NFC showed faster healing compared to Suprathel^®^ (commercial lactocapromer-based wound dressing) [53]. BC dressings are cheaper than conventional synthetic fiber dressings [232].

The surface properties of cellulose-based wound materials can be modified via several methods, such as cross-linking with silane-based reagents [233]. The material was investigated for femoral artery and liver injury models. Chemical modification with organosilane enabled the formation of a hydrophobic layer that stopped blood penetration (blood loss < 50%) and accelerated the process of blood clotting. It required a short time for hemostasis in both models [233]. The high surface charge of cellulose nanomaterials enhanced protein adsorption and could promote cell adhesion [162].

A dressing prepared from cellulose is usually transparent, allowing the evaluation of the wound treatment without removing or exchanging the dressing [51]. A cellulose-based membrane offers good adhesion to the moist wound surface due to the abundant hydroxyl groups present in the cellulose structure without significant allergic or inflammatory responses [51]. Cellulose-based dressings enable faster self-detachment than commercially available wound healing dressings [53]. They can be used for infected wounds [234]. Thymol-enriched BC hydrogel can be used to treat and heal third-degree burn injuries [235].

Among several cellulose types, BC-based membranes are widely used for wound healing. However, the extraction process produces a low yield and is expensive. Nanocellulose extracted from wood pulp fibers can be an alternative to BC. Cellulose-based dressings suffer from swelling behavior due to their high affinity toward water molecules. The surface properties should be improved to meet requirements for wound healing. BC shows no antimicrobial activity. Thus, antimicrobial agents are usually required.

**Table 2 ijms-23-05405-t002:** Cellulose-based materials for wound dressing.

Materials	Fabrication	Cellulose Source	Form	Study	Closure (%)	Time (days)	Bacteria	Efficiency	Ref.
BC–CTS	Immersing BC in chitosan followed by freeze-drying	*Acetobacter xylinum*	Membrane	In vivo	85	8	*E. coli* *S. aureus*	99.9%	[227]
Cellulose nanofibrils	Filtration technique	Birch pulp fibers	Membrane	In vitro In vivo Clinical studies		8–9	*P. aeruginosa*, *S. aureus*	None	[53]
Carboxylated CNF	1. Autoclaved using NaOH 2. TEMPO-mediated oxidation	*Pinus radiata* bleached kraft pulp fibers	Gels	In vitro In vivo		24 h		60%	[87]
SSD/BC	Impregnation of BC with SSD via ultrasonication	Commercial membrane	Membrane	In vitro			*P. aeruginosa*, *E. coli*, *S. aureus*	6.5 mm	[214]
BC	Cultured bacteria in Hestrin and Schramm (HS)	*Acetobacter xylinum*	Film	In vitro In vivo	90	24 h			[220]
T-GNF	1. Alkali treatment 2. TEMPO-mediated oxidation	*Ginger fibers*	Hydrogels	In vitro	67	72 h	*E. coli, S. aureus*	0	[221]
BACNF/QCR	1. Cation exchange 2. Freeze-drying	Brown algae	Sponge	In vitro In vivo	100	12 d		6 mm	[228]

## 6. Drug and Gene Delivery Using Cellulose-Based Materials

Cellulose-based materials have advanced drug delivery [61,236,237,238,239,240]. For multifunctional applications, they can be conjugated with nanomaterials such as magnetic nanoparticles (MNPs) [241]. Cellulose can encapsulate drugs [242]. The functional groups of carboxymethyl cellulose (CMC) enabled its modification with folate for the selective release of the anticancer agent 2,4-dihydroxy-5-fluorpyrimidin (5-FU) [243]. Cellulose’s surface modification with folic acid ensures selective cell uptake and binding via a folate-receptor-mediated cellular mechanism [244,245]. Cellulose serves as an effective carrier for delivering hydrophobic drugs such as docetaxel, paclitaxel (PTX), and etoposide [246].

Hydroxypropyl methylcellulose improved the cellular uptake of curcumin (CUR) to treat prostate cancer cells [247]. CUR-conjugated cellulose exhibited significant changes in apoptosis compared to CUR alone. Cellulose also showed the highest cellular uptake compared to other carriers, such as β-cyclodextrin (CD), poly(lactic-co-glycolic acid) (PLGA), MNPs, and dendrimers [247]. TOCNFs and MOFs such as zeolitic imidazolate frameworks (ZIF-8) and Material Institute of Lavoisier (MIL-101 (Fe)) were used for the drug delivery of CUR (Figure 10) [248]. TOCNF/ZIF-8 underwent material processing into a 3D network via 3D printing [248]. Cellulose enables the printing of MOF powder. It serves as a binder and template for MOFs during in situ synthesis. The materials can release the CUR drug at physiological pH (5.5) [248].

Cellulose-based materials were also used to deliver oligonucleotides such as siRNAs [249,250]. They can be considered non-viral vectors [251,252]. CNCs were modified with poly(2-dimethylamino)ethyl methacrylate) (PDMAEMA) via atom transfer radical polymerization (ATRP) for gene delivery of pDNA (Figure 11). Polymerization occurred via the formation of disulfide (SS) bonds, and the product was denoted as CNC-SS-PDs (Figure 11) [253]. The CNC-SS-PDs exhibited good transfection efficacy with low cytotoxicity (Figure 11) [253]. Cellulose-based materials are promising as non-viral vectors for gene delivery [251,254,255,256,257,258,259].

Cellulose-based materials offer several advantages for drug delivery. They can be used to release water-soluble and insoluble drugs and ionizable and hydrophobic drugs [246,260]. Thus, they enable the co-delivery of two drugs [261]. CNC hydrogels promoted locally targeted drug release [262] with sustainable properties [263]. Cellulose can be fabricated as capsules without gelatin [264]. It can be used for oral drug delivery [265]. Drug delivery using cellulose-based hydrogels can be simulated for pH- and temperature-responsive release.

## 7. Scaffolds for Skin, Bone, and Tissue Engineering

Nanocellulose-based materials offer several advantages as scaffolds for tissue engineering (Table 3) [266]. They show high biocompatibility, good water absorption, high water retention, high optical transparency, and good mechanical properties. They can be fabricated using comprehensive methods, including solvent casting, electrospinning, freeze-drying, and 3D printing [266]. Cellulose materials offer easy custom-made processes for tissue engineering of damaged tissues or organs. They can be optimized to ensure the required hierarchical structure, pore size, surface functional groups, and mechanical properties. Thus, cellulose-based materials are promising for engineering blood vessels, skin, and organs [266]. Scaffolds should meet several essential requirements, including high biocompatibility to mimic the natural extracellular matrix (ECM) of native tissue. They should also support cell growth, proliferation, and differentiation (Table 3).

Bacterial cellulose (BC) was used in tissue-engineered blood vessels (TEBV) [267]. BC-based TEBV is promising for caliber vascular grafts to reconstruct tissues associated with vascular diseases [267]. BC was used to fabricate artificial blood vessels that can be used for microsurgery [63]. The vessels were abbreviated as BASYC^®,^ referring to **BA**cterial **SY**nthesized **C**ellulose. The prepared vessels exhibited high mechanical strength in their wet form with high water retention. They also showed low roughness of the inner surface and provided complete “vitalization” in the rat. BASYC^®^ was proposed as an artificial blood vessel in microsurgery [63].

BC was reported as a temporary skin substitute [254] and a scaffold for cartilage tissue engineering [268]. It was used for the in vitro seeding of cells, such as L929 cells/human osteoblasts [269]; human adipose-derived stem cells (hASCs) [270]; human urine-derived stem cells (hUSCs) [271]; human keratinocytes (HaCats) [272]; human vein endothelial cells (HVECs) [273,274]; human umbilical vein endothelial cells (HUVECs) [275]; equine-derived bone marrow mesenchymal stem cells (EqMSCs) [276]; human embryonic kidney 293 cells (HEK) [277]; bovine smooth muscle cells (SMCs) [275]; endothelial cells (ECs) [278]; chondrocytes [268]; epidermal cells [214]; and mouse leukemic monocyte-macrophage cells (RAW 264.7, Table 3) [220]. It showed high biocompatibility and promoted the proliferation of cells such as SMCs, leading to ingrowth of size > 40 μm after two weeks of culture on BC pellicles [267]. The proliferation of hASCs on BC film showed a plateau phase after nine days, indicating a single layer on the film [270]. Thus, BC is promising for tissue engineering.

BC pellicles showed the formation of an exemplary network similar to a collagen network [267]. It formed porous scaffolds with different pore sizes using sterile paraffin particles of varying size ranges (90–150 μm, 150–300 μm, and 300–500 μm) [271]. Biomolecules such as alginate were used to prepare sponges for oral tissue regeneration [272]. It can also be shaped into tubes using tubular template materials such as PDMS (polydimethylsiloxane) [275].

Cellulose offers several advantages. The cellulose composite exhibits high cell compatibility, water uptake, and mechanical strength. Cellulose-based materials exhibit good water and phosphate buffer saline (PBS) buffer stability. BC enabled the growth of multipotent mesenchymal stem cells (MSCs) [276]. BC scaffolds improved the adhesion, proliferation, and differentiation of MSCs [268]. Thus, BC/MSC hydrogel can be used to construct musculoskeletal tissue. Composites of polyvinyl alcohol (PVA) and BC can be promising biomaterials for soft tissue replacement [279].

The proliferation rates of several cells such as HaCats and gingival fibroblasts (GFs) using different supports were ordered in the following order: tissue culture plastic (TCP) > BC > BC-alginate (BCA) > alginate [272]. This character depends on the cell type. BC showed significantly higher chondrocyte growth levels than TCP and calcium alginate [268]. Unmodified BC supported the proliferation of chondrocytes at 50% of the collagen type II substrate [268]. However, BC-based materials exhibited good mechanical properties [268].

The 3D printing of bioink consisting of NFC and alginate was reported [280]. NFC ensures high shear-thinning properties, while alginate enables fast cross-linking ability. The 3D printing of bioink can be used for cartilage structures, including a human ear and sheep meniscus (Figure 12). The method can be used to print human chondrocyte nanocellulose-based bioink. Cellulose bioink showed 73% and 86% high cell viability after 1 and 7 days, respectively [280].

**Table 3 ijms-23-05405-t003:** Applications of cellulose-based materials for tissue engineering.

Cellulose	Source	Form	Fabrication Method	Study Type	Cells	Time (d)	Evaluation Method	Comments	Ref.
BC	*Acetobacter xylinum*	Films	Shaken in a culture flask	In vitro In vivo	hASCs	9	Optical density (OD)		[270]
		Scaffold	Fermentation into sterile paraffin particles		hUSC	7	Histology	No effect of pore size	[271]
BC/alginate (BCA)		Sponge	Freeze-drying	In vitro	HaCat	2	MTT assay	30% alginate	[272]
BC		Scaffold	Culture on TCP	In vitro	EqMSCs	14	OD	The seeded cells were metabolically active	[276]
BC		Tubes	Culture on PDMS tubes	In vitro In vivo	SMCs	7		No signs of inflammation	[275]
BC-CMC	*Gluconacetobacter saccharivorans*	Gel	Agitation overnight at room temperature	In vitro	HEK	1	Optical microscope		[281]
BC	*Acetobacter xylinum*	Tubes	Fermentation in glass tubes using a silicone support	In vitro	ECs	28	F Fluorescence microscope		[278]
		Scaffold	Freeze-drying	In-vitro In-vivo	Chondrocyte	8			[268]

**Notes**: MTT (3-(4,5-dimethyl thiazolyl-2)-2,5-diphenyltetrazolium bromide) assay; *Acetobacter xylinum and Gluconacetobacter xylinus* are used for the same strains.

Cellulose has been used for bone regeneration. BC assisted the synthesis of hydroxyapatite (HAP) with calcium-deficient and crystallite particles of nanoscale size [282,283]. BC was oxidized before the formation of HAP to produce highly degradable materials. HAP/BC was used for bone regeneration using osseous tissue [282]. The material can be degraded under physiological conditions, i.e., pH and temperature of 7 and 37 °C, respectively. Thus, it was proposed to stimulate bone colonization [282]. A composite of HAP/BC was reported to treat non-critical bone defects in rats [284]. The bone defects were repaired entirely after 16 days [284].

A nanocomposite material of BC networks and HAP was prepared using a wet chemical precipitation method using aqueous calcium nitrate and di-ammonium phosphate salts [277]. The dispersion can be improved by adding CMC (1% *w/v*). CMC increased the pore size of BC by 47.8%. The prepared composite, i.e., BC/HAP/CMC, supported the growth of HEK cells [277].

BC scaffolds were used for the growth of EqMSCs to apply in bone and cartilage tissue engineering [276]. They can be fabricated via freeze-drying. They can be prepared with fiber diameters and pore sizes of 32.08 ± 10.85 nm and 254.16 ± 76.65 nm, respectively. They exhibited high cytocompatibility and supported the adhesion and proliferation of the cells. They maintained the differentiation of EqMSCs. EqMSC/BC scaffolds are promising for bone and cartilage regeneration tissue engineering, similar to those reported using tissue-culture-treated plastic (TCP) [276].

Cellulose materials have also been reported for other biomedical implants, such as artificial cornea [285], urinary conduits [271], neuronal regeneration [286], and endodontic treatment [287]. Cellulose and its composites are promising and can be implemented in other applications.

Cellulose-based materials offer several advantages for tissue engineering. They provided high biocompatibility [288] and self-healing properties [289]. Cellulose materials can be conjugated with polymers to improve their properties. A rigid composite of tannic acid (TA)/CNC (TA@CNC) was incorporated into poly(vinyl alcohol) (PVA)-borax networks [290]. TA@CNC/PVA-borax hydrogels offered high toughness with self-healing properties [290]. The structure of cellulose materials such as BC hydrogel enables the proliferation of HVECs horizontally and their migration vertically [273,274]. Cell penetration into BC hydrogels depends on oxygen availability inside the hydrogels.

## 8. Advantages and Disadvantages of Cellulose-Based Nanomaterials

Cellulose exhibits high biocompatibility and hemocompatibility with good biodegradability. Mesenchymal stem cells can grow on a cellulose membrane with viability higher than 95% [291]. However, cellulose materials show slow or no degradation in vivo and in vitro. This property makes them suitable for applications as a scaffold. The degradability of cellulose can be improved via chemical oxidation of the amorphous regions of BC using TEMPO-mediated oxidation [285,292], periodate [285], exposure to γ-radiation [293], or incorporation with an enzyme [294]. The chemical oxidation form 2,3-dialdehyde cellulose can easily degrade in vivo and in vitro. BC membranes can also be degraded via γ-radiation within 2–4 weeks [293].

The mechanical strength of nanomaterials can be improved using cellulose nanomaterials. Cellulose exhibits good mechanical properties [295]. Both air-dried and hot-pressed BC exhibit Young’s moduli higher than 18 GPa. Air-dried and hot-pressed BC show tensile strengths of >260 MPa and 216 MPa, respectively. Atomic force microscopy (AFM) was used to measure the mechanical properties of single BC nanofibers with diameters of 35–90 nm [296]. Data analysis revealed a high Young’s modulus of 78  ±  17 GPa [296]. BC-based hydrogel exhibited an elastic modulus of 1.7 MPa [297]. It possessed a very high water-holding capacity, even under pressures as high as 3.7 MPa. It recovered its original shape, even after repeated compression [297].

Cellulose retains moisture. The hydroxyl groups of cellulose material maintain the material’s hydrophilicity. The hydroxyl groups have high water capacity with good mechanical and thermal properties. These functional groups play an essential role in interfacial interactions during a composite [285].

BC offers protection from UV radiation and heavy metal stress. Cellulose exhibits resistance to aerobic environments.

The homogeneous distribution of cells on cellulose is vital for tissue engineering. The native BC membrane suffers from cell aggregation [298]. However, the modification of cellulose with RGD protein improved the distribution of cells [298]. The growth rate of muscle-derived cells can be increased by increasing the starch content [299]. The increase in the cell growth rate may be due to the porous nature of the composite [299].

## 9. Outlook

CNCs have been commercialized by several companies, such as Bio Vision (Canada), CelluForce (Canada; the trade name is *NCC*™), and US Forest Service Forest Products Laboratory (USA), while CNFs have been marketed by several European companies, such as Centre Technique du Papier (France)), Borregaard ChemCell (Norway), Innventia AB (Sweden), Stora Enso (Finland), and UPM fibril cellulose (Finland). Plant-based cellulose has been commercialized in several products, such as cellophane (transparent films), Rayon or Tencel™ (synthetic textile fibers), Bionext^®^ (commercial BC extracted from *Acetobacter xylinum*), Surgicel™, and Interceed^®^. BC-based materials have been commercialized in several products, such as Bioprocess^®^, BASYC^®^, Biofill^®^, XCell^®^, and Gengiflex^®^. Microbial-based cellulose is free of lignin and hemicelluloses, unlike cellulose extracted from plant sources. For example, bacterial cellulose (BC) exhibits higher purity and better crystallinity than plant-derived cellulose. Thus, BC is suitable for biomedical applications. However, microbial-based cellulose requires high-security precautions to avoid the presence of microbial species inside the extracted cellulose. Cellulose can be produced from cheap sources and waste materials [300,301,302]. However, most of the current technologies for cellulose fabrication at the nanoscale are still expensive, require tedious efforts, and lack large-scale production for industrial and biomedical applications. The output of nanocellulose with high purity and the absence of endotoxins is highly needed for biomedical applications.

Cellulose-based antimicrobial agents are promising for the fabrication of membranes [50], fabrics [152], textiles [303,304,305], and food packaging [306,307]. They possess durable antibacterial activity with tunable properties such as hydrophobicity [308]. They can be modified with different antibacterial agents with permanent antibacterial properties [309]. Nanoparticles exhibit undesirable toxicity. Thus, some precautions or post-synthetic treatments should be performed. For instance, sulfidation was proposed to transform Ag NPs into highly insoluble forms for minimal cytotoxicity [310].

Cellulose-based materials can serve as green cross-linkers [311] and flexible platforms [312] for tissue engineering. They can be modified via in situ and ex-situ procedures [313]. They can be processed into several forms using various methods. Most of the available techniques lack large-scale production and require other materials such as binders or modifiers. Further investigations are needed to develop cellulose materials that meet customized requirements at a minimal cost.

## Figures and Tables

**Figure 1 ijms-23-05405-f001:**
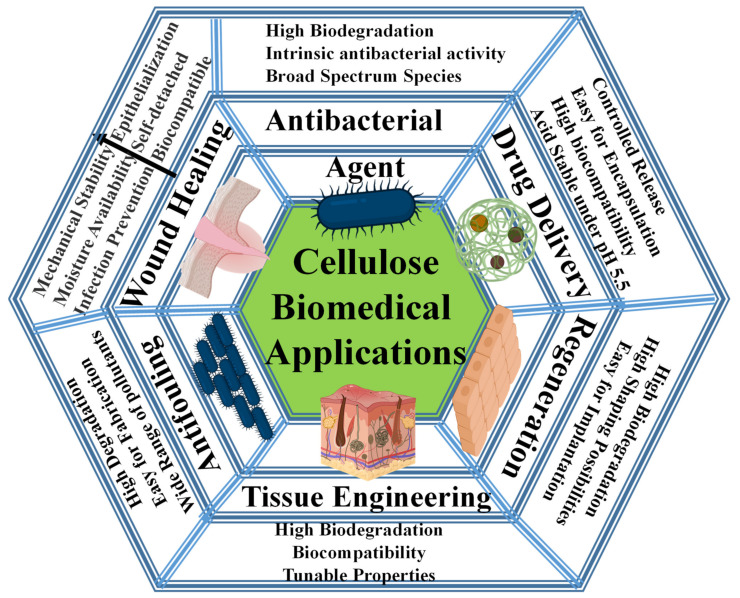
Overview of biomedical applications covered in this review.

**Figure 2 ijms-23-05405-f002:**
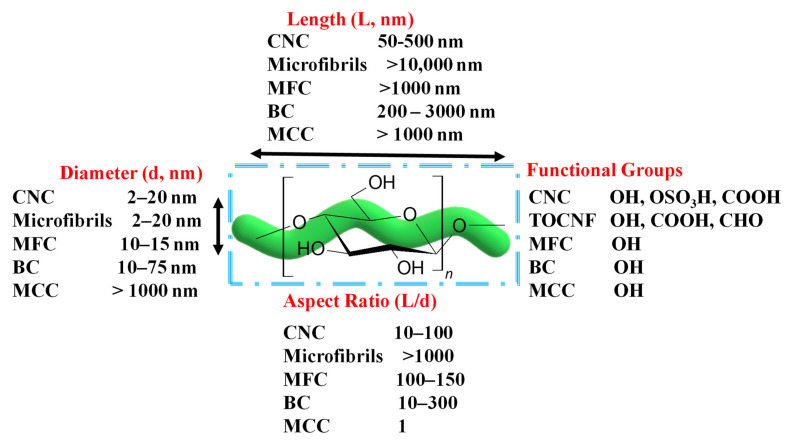
Summary of cellulose nanoparticles with sizes and functional groups.

**Figure 3 ijms-23-05405-f003:**
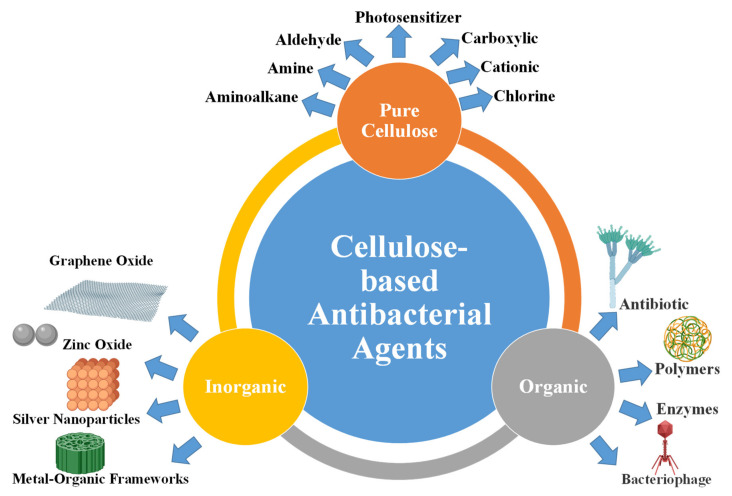
Cellulose-based materials as antibacterial agents.

**Figure 4 ijms-23-05405-f004:**
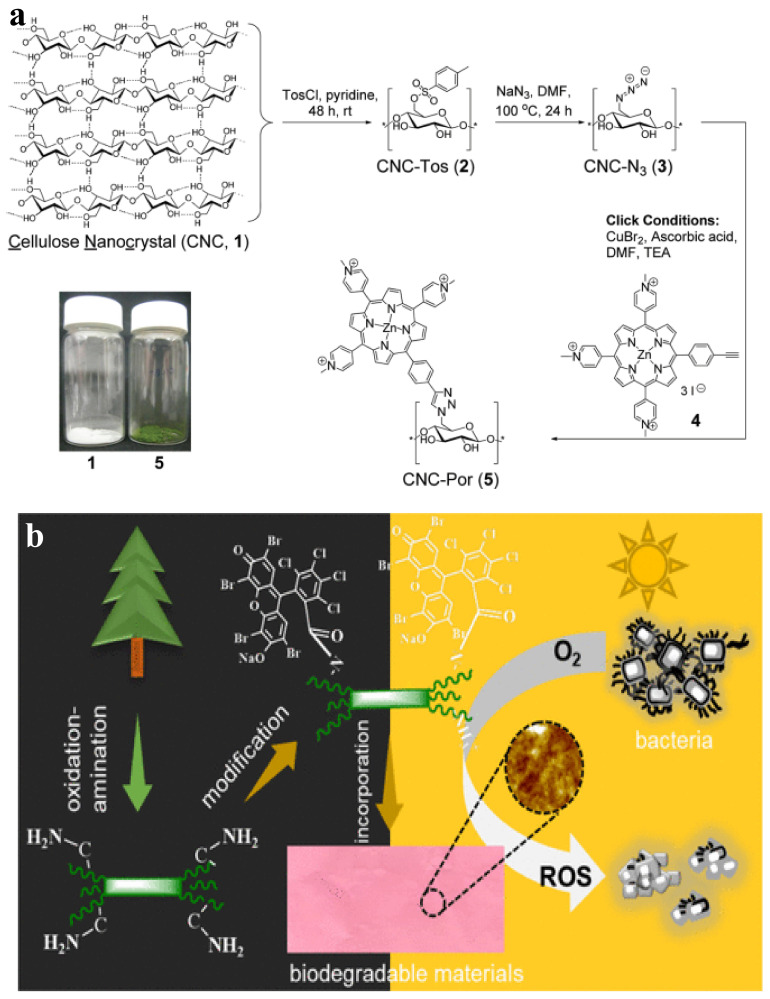
(**a**) Synthesis of CNC-Porphyrin; (1) CNC preparation via acid hydrolysis, (2) surface tosylation of CNC, CNC-Tos, (3) synthesis of azide-bearing CNC-N_3_, (4) click reaction of CNC-N_3_ with Porphyrin. Reprinted with permission from Ref. [96]. 2011, ACS (2011). (**b**) Chemical modification of ANCC with Rose Bengal as photosensitizer. Reprinted with permission from Ref. [97]. 2021, American Chemical Society (ACS).

**Figure 5 ijms-23-05405-f005:**
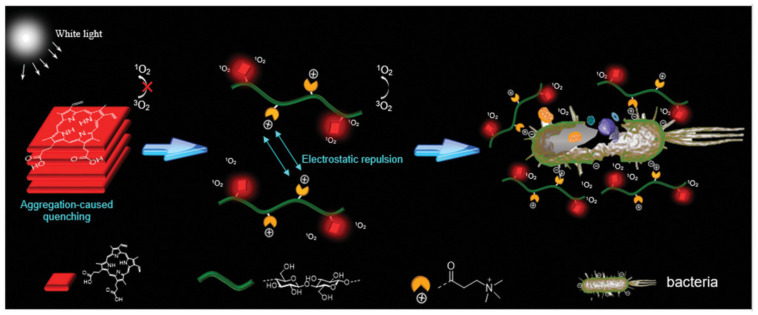
The antibacterial mechanism for porphyrin and quaternary ammonium-modified cellulose under light radiation. Reprinted with permission from Ref. [108]. 2019, John Wiley & Sons.

**Figure 6 ijms-23-05405-f006:**
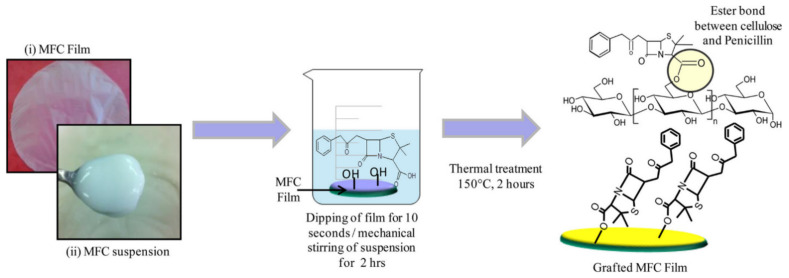
Schematic representation of suspension and film of MFC and chemical modification with Benzyl Penicillin via esterification. Reprinted with permission from Ref. [129]. 2015, ACS.

**Figure 7 ijms-23-05405-f007:**
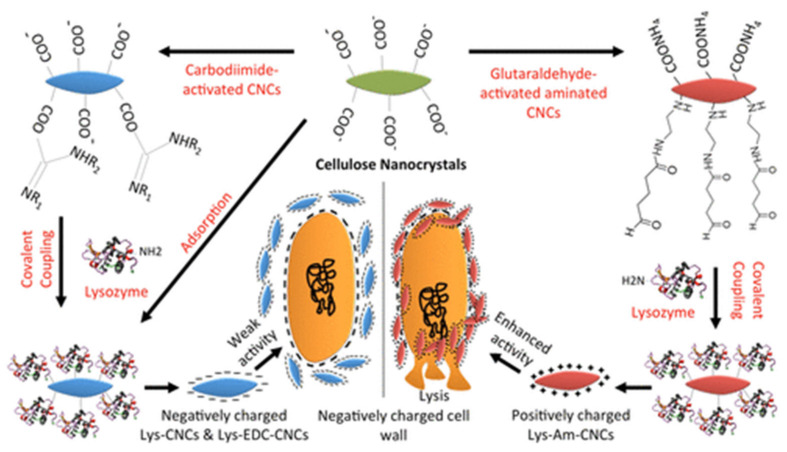
Schematic illustration of the immobilization of lysozymes on CNCs for antibacterial activity. Reprinted with permission from Ref. [168]. 2017, ACS.

**Figure 8 ijms-23-05405-f008:**
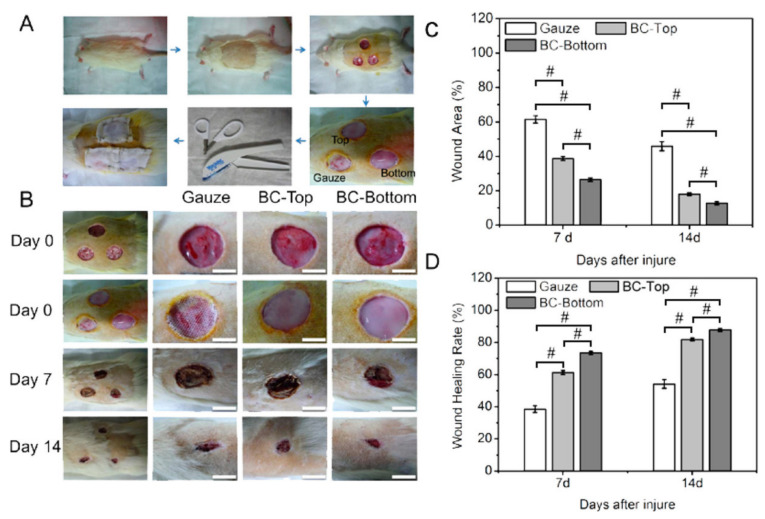
Wound healing treatment using BC-based dressing: (**A**) description of the operation on the skin injury model and the dynamic healing of a rat, (**B**) the progress (0–14 days) of healing for the skin injury model on Wistar rat using gauze and BC-based dressing of two sides, top and bottom (all scale bars equal 10 mm), (**C**) wound area progression after the injury and (**D**) wound healing rate. Error bars represent means ± standard deviation (SD) for n = 5 (^#^ *p* < 0.01).Reprinted with permission from Ref. [220]. 2015, American Chemical Society (ACS, 2015).

**Figure 9 ijms-23-05405-f009:**
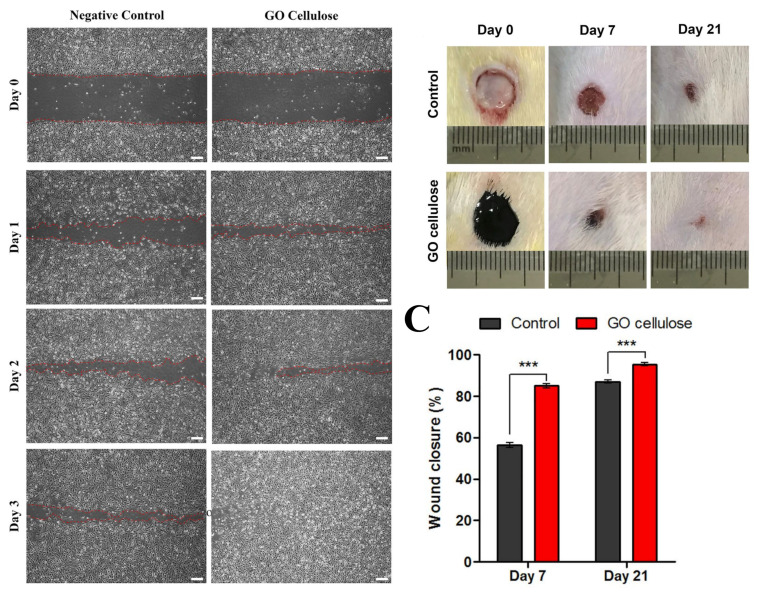
(**A**) Cell migration with and without GO–cellulose nanocomposite; red-dotted lines represent the wound edges, scale bar = 200 μm; (**B**) in vivo evaluation of the skin wounds of rats with and without GO–cellulose nanocomposite for post-wound induction on days 0, 7, and 21; and (**C**) the percentage of wound closure: significant differences were evaluated using one-way ANOVA, where *** *p* < 0.0001. Reprinted with permission from Ref. [230]. 2021, Elsevier.

**Figure 10 ijms-23-05405-f010:**
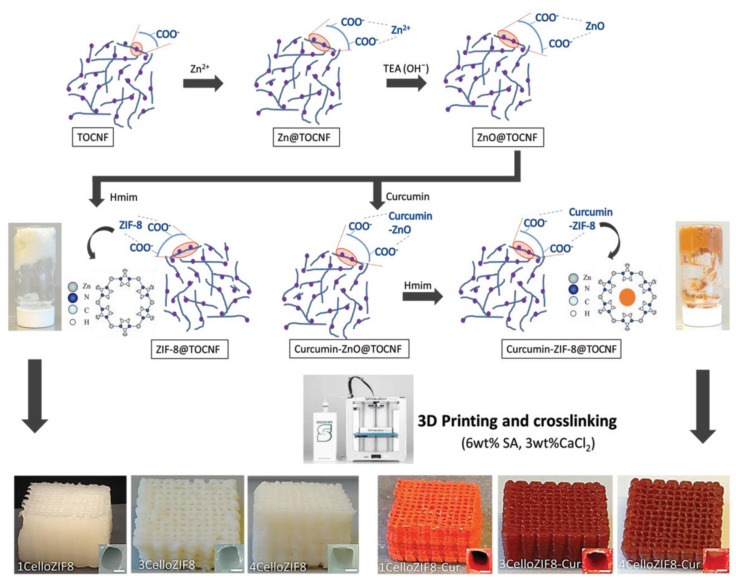
The synthesis procedure of cellulose-ZIF8 bioink and their processing into the 3D network via 3D printing. Reprinted with permission from Ref. [248]. 2019, John Wiley & Sons.

**Figure 11 ijms-23-05405-f011:**
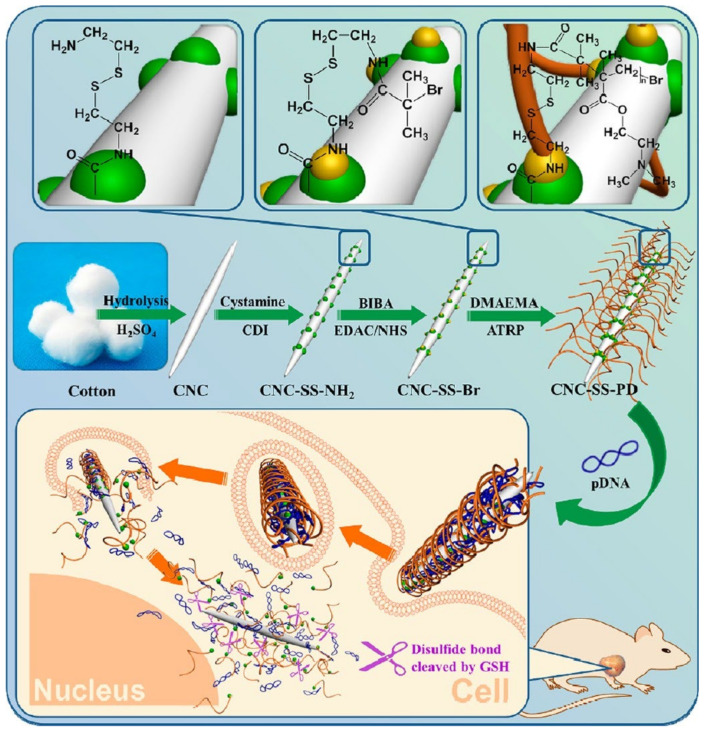
Schematic representation of the preparation of CNC-SS-PD and their use for gene delivery. Reprinted with permission from Ref. [253]. 2015, ACS.

**Figure 12 ijms-23-05405-f012:**
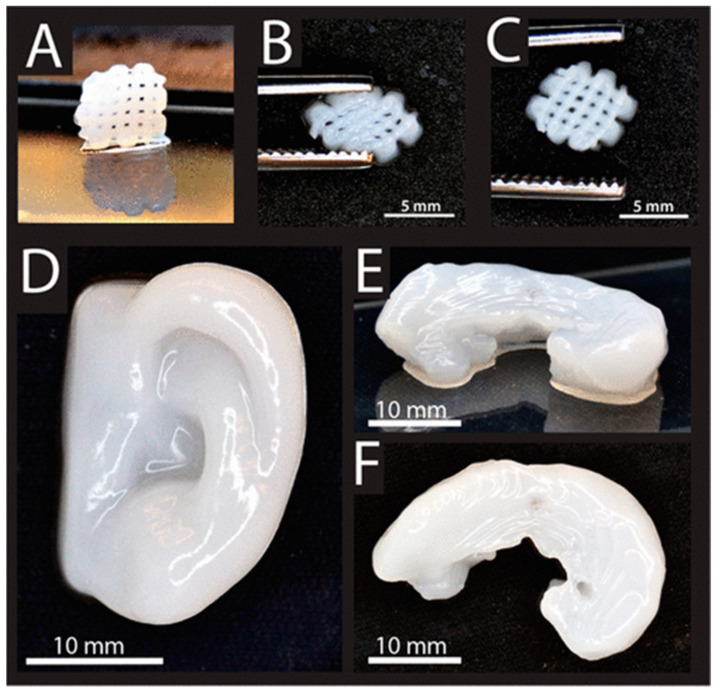
The 3D printing of NFC–alginate into (**A**) small grids (7.2 × 7.2 mm^2^), (**B**) after squeezing, and (**C**) restored after squeezing; (**D**–**F**) 3D-printed human ear in different views. Reprinted with permission from Ref. [280]. 2015, ACS.

## Data Availability

The data presented in this study are available on request from the corresponding author.

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
