# Peer review of "Cellulose-Based Nanomaterials Advance Biomedicine: A Review"

_ijms, 2022, doi:10.3390/ijms23105405_

Round 1
Reviewer 1 Report
Accept, the article has been improved well up to the reviewer's suggestions and comments
Author Response
Many thanks for the recommendation and all your valuable comments.
Reviewer 2 Report
The manuscript entitled „ A Review on Cellulose-based Materials for Biomedicine” by Hani Nasser Abdelhamid and Aji P. Mathew presents a review of cellulose-based compounds for use in biomedicine. Authors are citing almost 300 articles dedicated to the application of cellulose-based materials in biomedicine. The aim of the manuscript is very interesting and promising. However, some editorial recurring mistakes lower the overall grade of the manuscript, especially since they have not been corrected since the first version.
- What the Authors try to tell is in the following sentence: “They exhibit several properties, making them interested in biomedical application”?
- The same request to the Authors with this part of the manuscript: “Cellulose-based materials advanced drug delivery [55, 231–225]. They can be conjugated with nanomaterials such as magnetic nanoparticles (MNPs) to offer multifunctional applications [236]. They can be used for drug encapsulation [237]”
- Numerous repetitions of the same information, i.e. three times we can find more or less the same information: “Cellulose-inorganic hybrids exhibited high antibacterial performance. CNC/silver/polyamide membrane offered high antifouling with flux recovery of 92.6% using humic acid”.
- Sections 3.1 to 3.4 presents often the same information in different sections. It is confusing to a reader.
- The wrong number of figure 5.
- No information about nanocellulose and required dimensions in figure 2.
- What is a degree of crystallity other than cellulose-based nanomaterials?
- In the same part of the text, we can find the following sentences: “Cellulose-based materials have advanced biomedical applications [23-38]” and “Cellulose-based materials exhibit high potential for biomedicine applications [73]
Taking into account repeating mistakes, I cannot accept the manuscript in its present form.
Author Response
The manuscript entitled „ A Review on Cellulose-based Materials for Biomedicine” by Hani Nasser Abdelhamid and Aji P. Mathew presents a review of cellulose-based compounds for use in biomedicine. Authors are citing almost 300 articles dedicated to the application of cellulose-based materials in biomedicine. The aim of the manuscript is very interesting and promising. However, some editorial recurring mistakes lower the overall grade of the manuscript, especially since they have not been corrected since the first version.
- What the Authors try to tell is in the following sentence: “They exhibit several properties, making them interested in biomedical application”?
Response: The sentence has been revised, and additional text has been added accordingly.
- The same request to the Authors with this part of the manuscript: “Cellulose-based materials advanced drug delivery [55, 231–225]. They can be conjugated with nanomaterials such as magnetic nanoparticles (MNPs) to offer multifunctional applications [236]. They can be used for drug encapsulation [237]”
Response: Revise; thanks for the suggestion.
- Numerous repetitions of the same information, i.e. three times we can find more or less the same information: “Cellulose-inorganic hybrids exhibited high antibacterial performance. CNC/silver/polyamide membrane offered high antifouling with flux recovery of 92.6% using humic acid”.
Response: We ensure that there is no repetition. These two sentences show entirely different information. The first sentence establishes the antibacterial application. At the same time, sentences give information about the membrane, flux, and their application for antifouling.
- Sections 3.1 to 3.4 presents often the same information in different sections. It is confusing to a reader.
Response: The first section describes pure cellulose, while the second section represents organically modified cellulose.
- The wrong number of figure 5.
Response: Sorry for the typos. All numbers have been corrected.
- No information about nanocellulose and required dimensions in figure 2.
Response: All these derivatives and nanoparticles. The dimensions, as well as the aspect ratio, are included in Figure 2.
- What is a degree of crystallity other than cellulose-based nanomaterials?
Response: Cellulose nanomaterials exhibit a 50–90% degree of crystallinity.
- In the same part of the text, we can find the following sentences: “Cellulose-based materials have advanced biomedical applications [23-38]” and “Cellulose-based materials exhibit high potential for biomedicine applications [73]
Response: Revised accordingly.